# Effects of Low-Load Blood Flow Restriction Training on Hemodynamic Responses and Vascular Function in Older Adults: A Meta-Analysis

**DOI:** 10.3390/ijerph19116750

**Published:** 2022-05-31

**Authors:** Tianqi Zhang, Guixian Tian, Xing Wang

**Affiliations:** 1Department of Physical Education, Shanghai University of Political Science and Law, Shanghai 201701, China; ztq7999@163.com; 2Department of Business, Pingxiang College, Pingxiang 337055, China; tianguixian@zwu.edu.cn; 3Department of Physical Education and Training, Shanghai University of Sport, Shanghai 200438, China

**Keywords:** cardiovascular, older adults, blood flow restriction training, arterial stiffness

## Abstract

Background: The combination of low-load (LL) training with blood flow restriction (BFR) has recently been shown to trigger a series of hemodynamic responses and promote vascular function in various populations. To date, however, evidence is sparse as to how this training regimen influences hemodynamic response and vascular function in older adults. Objective: To systematically evaluate the effects of LL-BFR training on hemodynamic response and vascular function in older adults. Methods: A PRISMA-compliant systematic review and meta-analysis were conducted. The systematic literature research was performed in the following electronic databases from their inception to 30 February 2022: PubMed, Web of Science, Scopus, EBSCO host, the Cochrane Library and CNKI. Subsequently, a meta-analysis with inverse variance weighting was conducted. Results: A total of 1437 articles were screened, and 12 randomized controlled trials with a total 378 subjects were included in the meta-analysis. The meta-analysis results showed that LL-BFR training caused a significant acute increase in heart rate (WMD: 4.02, 95% CI: 0.93, 7.10, *p* < 0.05), systolic blood pressure (WMD: 5.05, 95% CI: 0.63, 9.48, *p* < 0.05) and diastolic blood pressure (WMD: 4.87, 95% CI: 1.37, 8.37, *p* < 0.01). The acute hemodynamic response induced by LL-BFR training is similar to that elicited by high-load (HL) training. Training volume, cuff pressure and width were identified as significant moderators in our subgroup and meta-regression analyses. After 30 min of training, resting systolic blood pressure significantly decreased (WMD: −6.595, 95% CI: −8.88, −3.31, *p* < 0.01) in the LL-BFR training group, but resting hemodynamic indexes exhibited no significant differences compared with common LL and HL training; long-term LL-BFR training resulted in significant improvements in flow-mediated vasodilation (FMD) (WMD: 1.30, 95% CI: 0.50, 2.10, *p* < 0.01), cardio ankle vascular index (CAVI) (WMD: 0.55, 95% CI: 0.11, 0.99, *p* < 0.05) and ankle brachial index (ABI) (WMD: 0.03, 95% CI: 0.00, 0.06, *p* < 0.05) in older adults. Conclusion: This systematic review and meta-analysis reveals that LL-BFR training will cause an acute hemodynamic response in older adults, which can return to normal levels 30 min after training, and systolic blood pressure significantly decreased. Furthermore, the beneficial effect of LL-BFR training on vascular function is to improve FMD, CAVI and ABI of older adults. However, due to the influence of the quality of the included studies and the sample size, more high-quality studies are needed to confirm such issues as BFR pressure and training risk.

## 1. Introduction

Aging is a risk factor for cardiovascular diseases and more than 90% of cardiovascular diseases occur in older adults over 55 years old [1]. According to the China cardiovascular disease report 2018, the number of cardiovascular disease deaths accounts for 40% of residents’ disease deaths, ranking first in China. Physical training has a positive effect on the cardiovascular health of older adults. The American Heart Association (AHA) regards physical exercise as one of the seven evaluation indexes of cardiovascular health [2], which can predict and reduce the risk of cardiovascular and cerebrovascular diseases, all-cause death and death from cardiovascular causes [3,4].

Blood flow restriction (BFR) training, also known as KAATSU training, involves placing a pressure band at the proximal end of an extremity to limit the blood flow of the distal muscle, resulting in increased muscle volume and strength. This procedure can be combined with low-load (LL) training. Compared with traditional training such as resistance training, LL-BFR training has the characteristics of low load and high benefit. Research has shown that only 20~30% 1-Repetition Maximum (1RM) BFR resistance training (RT) can produce the same level of benefits as high-load (HL) training, so it is more suitable for rehabilitation treatment groups or older adult groups [5]. It is worth noting that the muscle gain of BFR training is currently recognized, but its effect on cardiovascular health receives little attention. Because the ischemic and hypoxic training environment of LL-BFR training will cause higher levels of metabolic stress and nerve activation during training, this results in increased stress on the body's cardiovascular system. In view of the aging of the cardiovascular system in the elderly, which leads to changes in its function and structure, a reasonable measurement and evaluation of the degree of change in relevant indexes is an important aspect related to the safety of any intervention program utilizing LL-BFR training.

In traditional moderate/high-load training, with the increase in training load, a series of cardiovascular reactions, such as increased heart rate and blood pressure, will be caused. However, it is the repeated exposure to this change that produces positive functional and structural adaptions of blood vessels, including the effect on the function of endothelial cells and smooth muscle cells, as well as the structural remodeling of arteries. At present, research results relating to LL-BFR training are still controversial. Some studies have shown that LL-BFR training is not related to additional adverse cardiovascular events, and an acute and local increase in blood pressure caused by BFR training can produce a variety of positive cardiovascular adaptations, such as improved vascular endothelial function [6], peripheral blood circulation [7] and arterial and venous compliance [8,9,10]. However, some scholars believe that LL-BFR training may cause hyperactivity of the sympathetic nerve, leading to an acute increase in blood pressure and vascular resistance in the older adults [11], as well as vascular function damage related to hemorheology and shear stress, etc. [12,13]. In existing studies, discrepancies in the different training interventions and BFR pressure schemes may be an important reason for the inconsistent results. Because the relevant studies are relatively scattered, it is difficult to provide reliable evidence support for clinical practice.

In view of this, in this paper, the following issues will be explored on the basis of previous relevant studies: after adding new evidence, we will examine whether LL-BFR training causes an additional cardiovascular response and how the recovery effect is impacted compared to non-BFR training and control conditions; whether long-term LL-BFR training significantly improves vascular function, and whether new outcome indexes are added; whether there are differences in the effects of training on the cardiovascular system of older adults under different exercise interventions and BFR pressure intensities. In this study, we focus on the above problems so as to provide a reliable basis for accurately formulating BFR training prescriptions and improving the cardiovascular health of older adults.

## 2. Materials and Methods

This study was conducted in accordance with the PRISMA statement (preferred reporting items for systematic reviews and meta-analyses) and the requirements of the Cochrane Collaboration Workbook [14].

### 2.1. Literature Retrieval Strategy

In the English databases of PubMed, Web of Science, EBSCO host, the Cochrane Library, Chinese databases CNKI and Wan Fang data, the data on the randomized controlled experiment (RCT) of blood flow restriction training on lower limb muscle strength and motor function in older adults were retrieved online and the retrieval time was from the establishment of a database to 30 February 2022. In addition, the relevant literature was supplemented by tracking the relevant systematic review and references included in the literature, and manually searching the journals related to the topic. In the search, a combination of MeSH and free terms was used. With PubMed taken as an example, the specific retrieval strategies were as follows:

#1 aged [Mesh] OR elderly [Title/Abstract] OR “older adult” [Title/Abstract] OR “old people” [Title/Abstract] OR “Postmenopausal women” [Title/Abstract]

#2 “blood flow restriction” [Title/Abstract] OR KAATSU [Title/Abstract] OR “occlusion training” [Title/Abstract] OR “vascular occlusion” [Title/Abstract] OR ischemia [Title/Abstract]

#3 “Neurovascular Coupling” [Mesh] OR “hemodynamic response” [Title/Abstract] OR “Blood pressure” [Title/Abstract] OR “Heart rate” [Title/Abstract]

#4 “vascular function” [Mesh] OR “endothelial function” [Title/Abstract] OR “vascular compliance” [Title/Abstract] OR “Vascular Stiffnesses” [Title/Abstract] OR “vascular resistance” [Title/Abstract] OR “flow-mediated dilatation” [Title/Abstract] OR “Cardio Ankle Vascular Index”.

#5 randomized controlled trial [Publication Type] OR randomized [Title/Abstract] OR controlled [Title/Abstract] OR trial [Title/Abstract]

#6 #1 AND #2 AND #3 AND #4 AND #5

### 2.2. Inclusion and Exclusion Criteria

#### 2.2.1. Inclusion Criteria

(1)Research Type: Randomized controlled trial (RCT).(2)Research Subject: ① Age ≥ 50 years old, any gender; ② No mental disorders, musculoskeletal diseases or serious cardiovascular risk.(3)Intervention Method: ① at least one experimental group shall use a training intervention based on blood flow restriction or vascular BFR pressure under any condition of cycle, duration, intensity and/or frequency; blood flow restriction training refers to LL-training with BFR pressure of the proximal limb, which includes LL-walking training (WT) and LL-resistance training (RT) combined with BFR. ② No BFR was implemented in the control group, and LL-training (<50% 1RM), HL-training (≥65% 1RM) and daily activities were included in the control group; ③ If multiple BFR training groups or control group are reported in the same literature, they shall be analyzed separately.(4)Outcome Indicator: The primary outcome indicators of the included trials were acute (measured immediately after training) and resting (measured ≥ 30 min after training) hemodynamic response; the secondary outcome indicators were vascular function including flow-mediated vasodilation (FMD), cardio ankle vascular index (CAVI) ankle brachial index (ABI) and venous compliance (VC).

#### 2.2.2. Exclusion Criteria

(1) Non-randomized controlled trial; (2) Overview, review, animal experiments, repeatedly published literature, etc.; (3) The outcome indexes do not meet the requirements, the data are incomplete or the data cannot be transformed; (4) The study that the experimental group participated in is a non-BFR training intervention, such as a low/high-intensity non-BFR training intervention; (5) Non-Chinese and English Literature.

### 2.3. Literature Screening, Data Extraction and Quality Evaluation

#### 2.3.1. Literature Screening and Data Extraction

The two researchers screened the literature, extracted the data and reviewed each other in an independent double-blind manner according to the inclusion and exclusion criteria. If there were differences in literature inclusion and data processing, etc., the third author participated in the discussion and a joint decision was reached. The extracted data were as follows: ① basic information (author, publication date, country, sample size, age); ② BFR training protocol, treatment of control group and outcome indexes.

#### 2.3.2. Quality Evaluation

RevMan 5.4 statistical software was used to assess the methodological quality of the included literature based on the Cochrane Collaboration's tool for assessing risk of bias. The main contents include the following: (1) Selection bias (random sequence generation, allocation concealment); (2) Performance bias (blinding of participants and personal); (3) Detection bias (blinding of outcome assessment); (4) Attrition bias (incomplete outcome data); (5) Reporting bias (selective reporting); (6) Other bias (other factors causing the risk of bias). The quality is scored according to three levels (low risk, high risk and unclear risk). The quality evaluation was carried out by two researchers independently and then cross-checked. In cases of disagreement, a consensus was reached through discussion or reference to third-party opinions. All 7 items met the standard of low risk of bias; some met the standard of moderate risk of bias; none of the seven items met the standard of high risk of bias.

### 2.4. Statistical Analysis

The data processing software of Stata 14.0 was used to perform the meta-analysis in strict accordance with PRISMA guidelines. *p* value and *I*^2^ were used for the test for heterogeneity. If there was statistical heterogeneity among the research results (*I*^2^ ≥ 50%, *p* < 0.10), the random effect model was selected, otherwise, the fixed effect model was used. The processing data were continuous data, and the Mean and standard deviation (SD) were calculated using the change value from pre-training to post-training within each group. If the SD was not provided, it was calculated by transforming the standard error (SE), confidence interval (CI) and *t*-value. WMD was used to measure the effect size when the measurement methods and measurement units of outcome indexes of studies were the same, otherwise SMD was used. When *p* < 0.05, there was a significant difference between the experimental and control groups, proving that the meta-analysis results were statistically significant. Stata 14.0 was used for sensitivity analysis on all outcome indexes included in the literature. The effect size of the research intervention was used as the dependent variable, and the factors that may affect the heterogeneity of the meta-analysis (training cycle, volume and different condition of control group) as the covariate; the restricted maximum likelihood method (REML) was used for meta-regression analysis.

## 3. Results

### 3.1. Literature Search Results

By searching PubMed, Web of Science, the Cochrane Library, EMBASE, CBM, CNKI, VIP and Wan Fang data, a total of 1437 literature were retrieved. No literature was obtained from other resources in this paper. After deduplication, reading topics and abstracts, full-text re-screening and exclusion of unqualified literature, 12 publications were finally included in the meta-analysis, all retrieved from English databases. The literature screening process is shown as follows (Figure 1).

### 3.2. Basic Characteristics of Literature Inclusion

This study includes 12 RCT studies [6,8,9,15,16,17,18,19,20,21,22,23], with a total sample size of 378, 169 in the experimental group and 209 in the control group. Table 1 shows the basic information, such as training intervention characteristics, BFR approach and outcome indicators included in the study.

### 3.3. Quality Evaluation of Included Literature

The RCT bias risk assessment tool in the Cochrane Collaboration was used to evaluate the methodological quality of a single RCT (Figure 2). In the 12 studies included in this research, the experiments performed were randomized controlled trials, of which nine described the method of random sequence generation. The methods of random allocation concealment were described in 1 publication. The method of blinding was not carried out on subjects and researchers in all studies; in four studies, the blinding method for the outcome evaluator was described, and in the other studies the blinding method was unclear. In two studies, the number of people lost to follow-up and data processing were reported in detail. In all 12 studies complete data were reported. 

### 3.4. Acute Hemodynamic Response

#### 3.4.1. Heart Rate (HR)

In 11 trials, the acute changes in HR were compared between the LL-BFR training group and the control group. The results showed that the acute effects of LL-BFR training on HR were significantly higher than for the non-BFR training group (WMD = 4.02, 95% CI: [0.93, 7.10], *p* < 0.05) with high heterogeneity (*I*^2^ = 86.3%, *p* < 0.10, Figure 3).

#### 3.4.2. Systolic Blood Pressure (SBP)

In 11 trials, the acute changes in SBP were compared between the LL-BFR training group and the control group. There was heterogeneity between trials (*I*^2^ = 90.0%, *p* < 0.10), thus the random effect model was used. The results indicated that SBP increased in the LL-BFR group, and that this increase was significantly higher than for the control group (WMD = 5.05, 95% CI: [0.63, 9.48], *p* < 0.05, Figure 4).

#### 3.4.3. Diastolic Blood Pressure (DBP)

In our analysis, we also assessed any acute changes in DBP after LL-BFR training. Heterogeneity existed between trials (*I*^2^ = 93.5%, *p* < 0.10), so the random effect model was used. The results of the meta-analysis showed that the acute changes in DBP after LL-BFR training were significantly higher than those in the control group (WMD = 4.87, 95% CI: [1.37, 8.37], *p* < 0.01, Figure 5).

#### 3.4.4. Meta-Regression and Subgroup Analysis

A meta-regression analysis of 30 trials was performed to explore the heterogeneity in acute hemodynamic response outcomes (Table 2). The results show that exercise volume (*p* < 0.05, 95% CI = −0.09, −0.01), BFR cuff pressure (*p* < 0.05, 95% CI: [−0.05, −0.01]) and cuff width (*p* < 0.05, 95% CI: [−0.34, −0.04]) were significantly and negatively correlated with outcomes. Differences in the treatment of control groups were not significantly related to outcomes.

To assess LL-BFR training protocols for older adults, subgroup analysis was conducted to compare the effects of BFR-RT with those of low-load resistance training (LL-RT) and high-load resistance training (HL-RT). The results (Table 3) showed that, in comparison with LL-RT, BFR-RT resulted in a significant additional increase in HR, SBP and DBP, whose increasing levels were similar to those after HL-RT.

Stratification of different BFR-RT protocols according to regulatory variables followed by subgroup analysis revealed the following: in a training protocol of 45 repetitions in a single session with a cuff pressure of 190~200 mmHg and cuff width of 18 cm, there was no significant difference in the acute cardiovascular responses elicited by BFR-RT and LL-RT, and there was no heterogeneity between studies, which was the only difference from the above findings.

#### 3.4.5. Sensitivity Analysis for Acute Hemodynamic Response Outcome

Sensitivity analysis showed that the point estimates of effect sizes were within the 95% CI of the combined effect size. Excluding a trial had a small effect on the effect size of the working memory index, indicating that the meta-analysis results were stable (Figure 6).

### 3.5. Resting Hemodynamic after LL-BFR Training

#### 3.5.1. Heart Rate

In eight trials, the changes in resting HR (measured ≥ 30 min after training) were compared between training with and without BFR. There was no significant change in resting HR between LL-BFR and non-BFR training (WMD = 0.816, 95% CI: [−0.85, 2.48], *p* = 0.337) with moderate heterogeneity (*I*^2^ = 34.4 %, *p* > 0.09, Figure 7).

#### 3.5.2. Systolic Blood Pressure (SBP)

In nine trials, the changes in resting SBP (measured ≥ 30 min after training) were compared between training with and without BFR. There was moderate heterogeneity between trials (*I*^2^ = 45.9%, *p* > 0.09), so the fixed effect model was used for analysis. The results showed that resting SBP decreased significantly after LL-BFR training compared with the control group (WMD = −6.595, 95% CI: [−8.88, −3.31], *p* < 0.01, Figure 8).

#### 3.5.3. Diastolic Blood Pressure (DBP)

In nine trials, the changes in resting DBP (measured ≥ 30 min after training) were compared between training with and without BFR. There were no significant differences in the changes in DBP between BFR and non-BFR training conditions (WMD = −0.119, 95% CI: [1.55, 1.32], *p* < 0.01), and no heterogeneity (*I*^2^ = 0%, *p* > 0.09, Figure 9).

#### 3.5.4. Meta-Regression and Subgroup Analysis

Meta-regression and subgroup analysis were also performed to explore the potential effect of a regulator variable on resting hemodynamic outcomes. Based on the results of the meta-regression (Table 4), the effect sizes did not significantly vary according to exercise cycle, cuff pressure, cuff width or differences in control groups (all *p* ≥ 0.06).

The results of the subgroup analysis are shown in Table 5. Compared with LL-RT, BFR-RT has similar effects on resting HR, with an additional reduction in resting SBP and DBP, but the effects did not reach statistical significance. Interestingly, when LL-RT and BFR-RT lasted 8 weeks with a cuff pressure of 150~170 mmhg and a cuff width of 23 cm, the resting hemodynamic response after BFR-RT decreased significantly compared with that after LL-RT.

In addition, the results of the subgroup analysis showed that there was no significant difference in the effect of BFR-RT and HL-RT on resting hemodynamics, and different training protocols did not exert an additional effect on this result.

#### 3.5.5. Sensitivity Analysis for Resting Hemodynamic Outcome

Sensitivity analysis showed that the point estimates of effect sizes were within the 95% CI of the combined effect size. Excluding a trial had a small effect on the effect size of the working memory index, indicating that the meta-analysis results were stable (Figure 10).

### 3.6. Vascular Function

#### 3.6.1. Flow Mediated Vasodilation (FMD)

Four studies were included to evaluate the effect of LL-BFR training on FMD intervention in the older adults. There was no heterogeneity between the studies (*I*^2^ = 0%, *p* > 0.09), so the fixed effect model was used for meta-analysis. The results indicated that the effect of the intervention in the LL-BFR group was greater than for the control group (WMD = 1.30, 95% CI: [0.50, 2.10], *p* < 0.01, Figure 11).

#### 3.6.2. Cardio Ankle Vascular Index (CAVI)

Four studies were included to evaluate the effect of LL-BFR training on CAVI intervention in the older adults. There was no heterogeneity between the studies (*I*^2^ = 0%, *p* > 0.01), so the fixed effect model was used for meta-analysis. The pooled effect size was WMD = 0.55, 95% CI: [0.11, 0.99], *p* < 0.05, indicating that the effect on CAVI of the older adults in the LL-BFR group was significantly improved compared with the control group (Figure 12).

#### 3.6.3. Ankle Brachial Index (ABI)

A total of four studies were included to evaluate the effect of LL-BFR training on ABI intervention in the older adults. There was no heterogeneity between studies (*I*^2^ = 0%, *p* > 0.09), so a fixed-effects model was used for meta-analysis. The pooled effect size was WMD = 0.03, 95% CI: [0.00, 0.06], *p* < 0.05, indicating that the effect of LL-BFR training on the ABI in the older adults was significantly improved compared with the control group (Figure 13).

#### 3.6.4. Venous Compliance (VC)

Two studies were included to evaluate the effect of LL-BFR training on VC in the older adults. There were no statistical differences between LL-BFR and control group for VC intervention of the older adults (WMD = 0.00, 95% CI (−0.00, 0.01), *p* > 0.06) with no heterogeneity (*I*^2^ = 0%, *p* > 0.10, Figure 14).

## 4. Discussion

We conducted an updated meta-analysis of RCTs of LL-BFR training for hemodynamic response and vascular function of older adults. A total of 13 studies of LL-BFR training were included, involving 73 comparisons with the non-training group or conventional LL and HL training groups. In addition, meta-regression and subgroup analyses were performed to investigate the potential sources of heterogeneity.

The results have shown that LL-BFR training induced acute hemodynamic responses, including increases in heart rate, systolic and diastolic blood pressure in older adults. Their increasing levels were similar to those observed for conventional HL training. Relevant studies have suggested that BFR training works by aggravating the stimulation of the exercise pressor reflex (EPR) [11], a reflex that significantly contributes to the autonomic cardiovascular response to exercise. During LL-BFR training, local muscles are in a state of relative ischemia and hypoxia through cuff pressure and insufficient blood supply, leading to an increase in metabolites, which activates chemical receptors in skeletal muscle to induce sympathetic activity and reduce parasympathetic activity [24,25,26]. Meanwhile, after measurement, the release of norepinephrine (NE) after LL-BFR training was significantly higher than that in the non-BFR training group. NE leads to the temporary contraction of peripheral blood vessels, resulting in an increase in peripheral resistance [6], which will also lead to an increase in systolic and diastolic blood pressure during training. Based on the results of our meta-regression and subgroup analyses, the adjustment of training volume, BFR cuff pressure and width may have a significant effect in reducing acute cardiovascular responses. The protocol of resistance training 45 times with a BFR cuff pressure of 190–200 mmHg and a cuff width of 18 cm resulted in an increase in HR, SBP and DBP, similar to that caused by conventional LL-RT. However, as it is limited by the sample size of our subgroup analysis, this result still needs to be further investigated in diverse implementation settings of LL-BFR training.

The acute response of HR and blood pressure (BP) caused by LL-BFR training can recover 30 min after training, and the resting SBP was significantly lower than that observed for the control groups. A possible explanation for this positive result is that after the blood flow is released from the training muscle, the reperfusion process of locally and rapidly congested blood induces the release of endothelium-dependent vasodilator, resulting in an increase in local blood flow, thereby reducing vascular resistance and blood pressure [27,28]. Given that BFR training is conducive to reducing cardiac preload during training, some studies have shown that it can be used as a more effective exercise intervention in hypertensive patients [29,30]. However, in contrast with the results of meta-analyses conducted by previous researchers [31], in this study we noted that, while LL-BFR had a better hypotensive effect than the conventional LL-RT or HL-RT training, there was no statistically significant difference between them. Based on the fact that the adaptation of regional vessels to LL-BFR training takes time, the initially increased level of endothelium-mediated vasodilation can return to the baseline level with arterial remodeling induced by long-term training [32]. Therefore, the training cycle may have an impact on the degree of blood pressure change. It remains to be elucidated whether BFR training has a stable post-exercise hypotensive effect and what its mechanism of action might be.

The results of this study have shown that LL-BFR training has a stable effect on improving FMD in older adults. FMD enhancement is a beneficial factor for cardiovascular health, which is conducive to the improvement and prevention of atherosclerosis. The mechanism of FMD improvement is related to that of endothelial function, and relevant studies have found that hypoxia stress activated VEGF is the main reason for the improvement of FMD by BFR training [33]. Local ischemia-reperfusion during LL-BFR training can accelerate the release of VEGF in vascular endothelium and skeletal muscle cells and also cause an increase in the serum GH concentration [6,34], both of which processes can up-regulate the expression of nitric oxide synthase (enos-1) [35], which, combined with the protective effect of myocardial ischemic preconditioning [36], can adjust NO bioavailability and improve endothelial function [37,38].

CAVI and ABI are the two most commonly used parameters to evaluate arterial function from different angles and our research results have shown that 12 weeks of LL-BFR training has a significant positive effect on them. These findings are partially consistent with Liu [10], in which six RCTs were included to investigate the effects of BFR-RT on arterial compliance in a mixed-age population. Based on the above results, we believe that LL-RT combined with BFR can cause an increase in the acute hemodynamic response at a higher level compared with LL-RT, and the acute hemodynamic response shows a greater tendency to decline after recovery. Therefore, long-term blood flow restriction training may cause changes in arterial load-bearing properties and adaptive changes in arterial structures in older adults. Meanwhile, due to the relatively low oxidative stress [39] and inflammatory responses [40] induced by LL-BFR training, BFR training is relatively beneficial to arterial function in older adults.

Decreased venous compliance is one of the risk factors for varicose veins and deep venous thrombosis [41]. At present, there are few studies that measure the effect of LL-BFR training on the venous compliance of older adults, and their results remain unclear. From the current research results, the beneficial effect of LL-BFR combined with walking training on venous function in older adults is greater than that of resistance training. The calf muscles are used more in treadmill walking training than in resistance training, resulting in a significant increase in local muscle activation and blood flow in the calf, which is a beneficial factor for improving venous compliance [9,42]. Compared with the improvement effect of general aerobic exercise [43,44], in BFR walking, when physical activity is in progress, venous outflow and arterial inflow are restricted, causing venous blood accumulation in the distal limbs by the application of pressure to the trained muscles. The blood stagnation in lower limbs leads to changes in the hydrostatic pressure in the legs, thus affecting cardiovascular reflex responses. These changes may synergistically affect the venous vascular function and improve venous compliance in the elderly in a shorter period of time. In future studies, it is still necessary to further study the valve state, hydrostatic force and cardiovascular reflex responses in more detail to clarify their mechanisms.

## 5. Research Limitations

There are some limitations and deficiencies in this study. First, some outcome categories or subgroups included data from a small number of trials, rendering resultant effect sizes potentially uncertain. Second, since few studies on vascular function indexes were included, no subgroup analysis was performed on these. Therefore, the role of regulatory variables cannot be defined. Third, this study did not consider the potential influence of BMI and other personal health factors on the results. The transient rise and instability of arterial blood pressure caused by sympathetic hyperactivity are risk factors for the occurrence of vascular diseases in older adults [45,46]. In order to avoid the possible risk of blind LL-BFR training, a large sample size is still needed to verify the regulatory effect of more variables and the appropriate BFR training approach for older adults. Fourth, at present, the mechanism of blood pressure changes in older adults after LL-BFR training remains to be explored, and could be considered further from the perspective of autonomic nerve regulation.

## 6. Conclusions

This meta-analysis has shown that LL-BFR training will cause an acute increase in heart rate and blood pressure in older adults. The level of acute hemodynamic responses induced by LL-BFR training is significantly higher than that elicited by LL training and similar to that caused by HL training. In addition, training volume, cuff pressure and cuff width were identified as significant moderators in our subgroup and meta-regression analyses. LL-BFR training results in a significant decrease in SBP after recovery, and there is no significant difference in resting hemodynamic response after recovery compared with regular resistance training.

The beneficial effect of LL-BFR training on vascular function in older adults lies in the improvement of FMD, CAVI and ABI. We identified no additional effect on venous compliance in the older adults undergoing BFR training compared with non-BFR groups.

## Figures and Tables

**Figure 1 ijerph-19-06750-f001:**
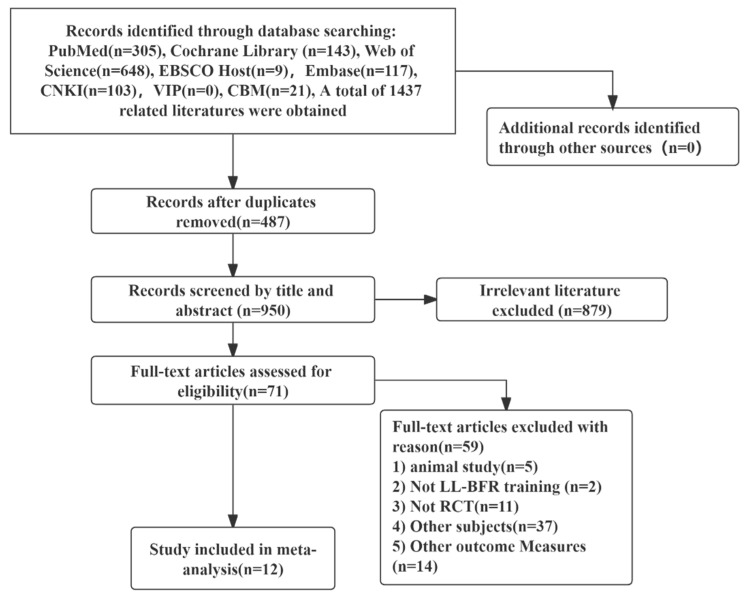
PRISMA flow diagram of the study selection process.

**Figure 2 ijerph-19-06750-f002:**
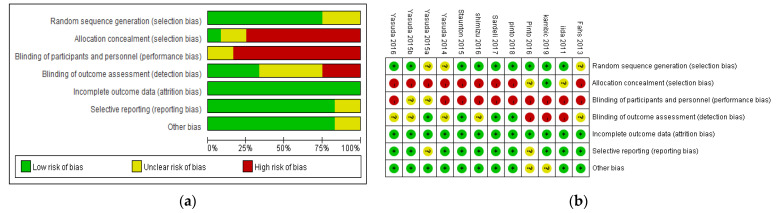
(**a**) Risk of bias graph; (**b**) Risk of bias summary.

**Figure 3 ijerph-19-06750-f003:**
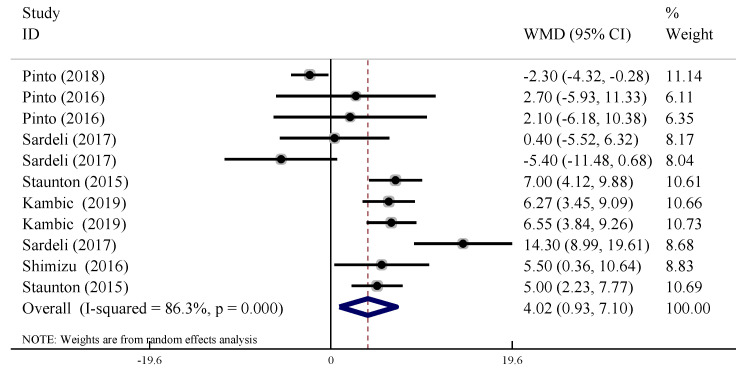
Acute effect of BFR-LI training on heart rate.

**Figure 4 ijerph-19-06750-f004:**
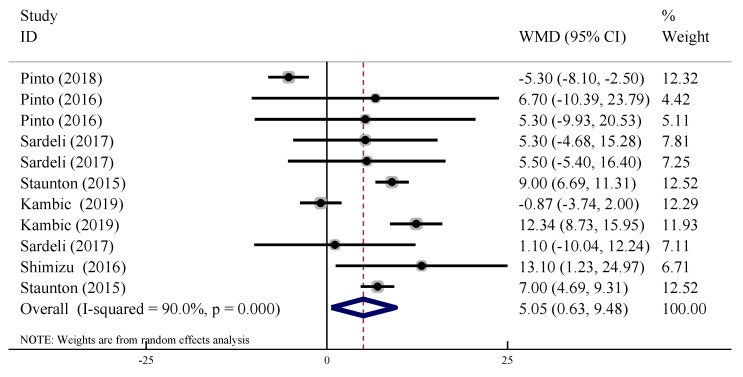
Forest plot of the acute effect of BFR-LI training on SBP.

**Figure 5 ijerph-19-06750-f005:**
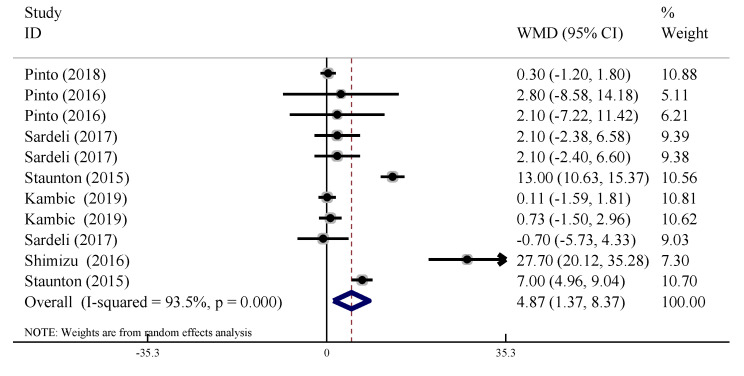
Forest plot of the acute effect of LL-BFR training on DBP.

**Figure 6 ijerph-19-06750-f006:**
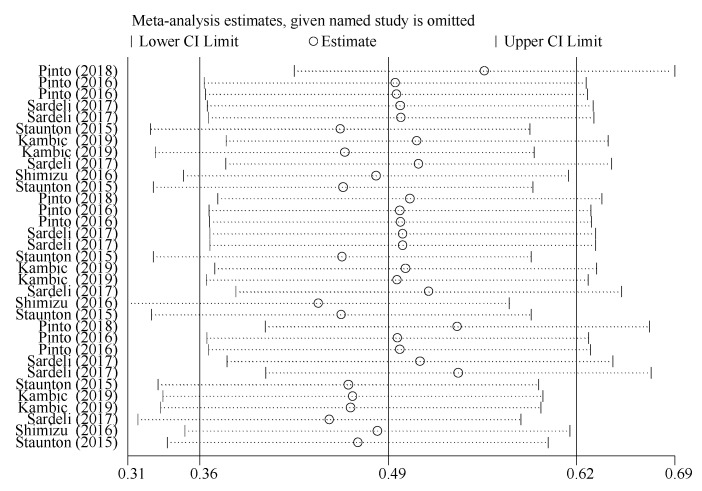
Sensitivity analysis for acute hemodynamic response outcomes.

**Figure 7 ijerph-19-06750-f007:**
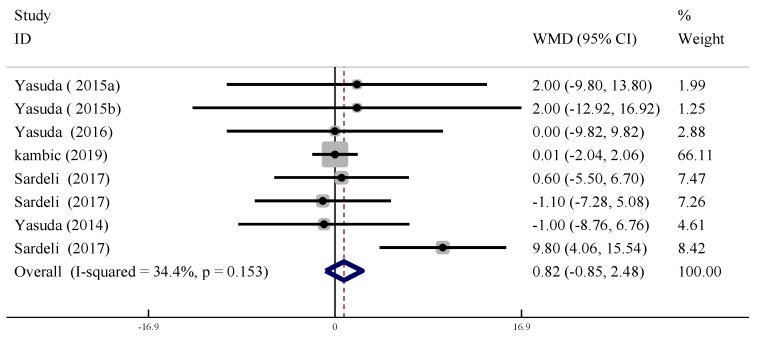
Forest plot of the effect of LL-BFR training on resting heart rate.

**Figure 8 ijerph-19-06750-f008:**
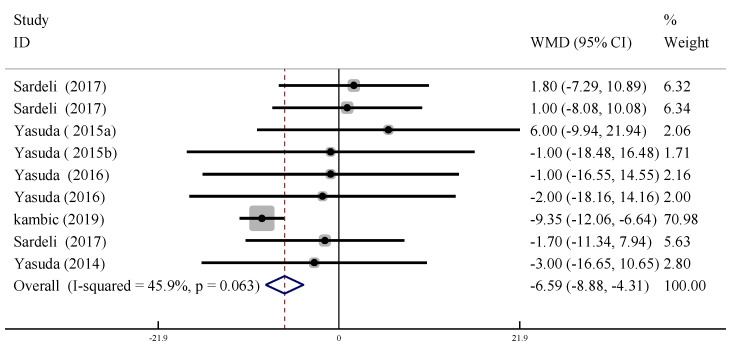
Forest plot of the effect of LL-BFR training on resting SBP.

**Figure 9 ijerph-19-06750-f009:**
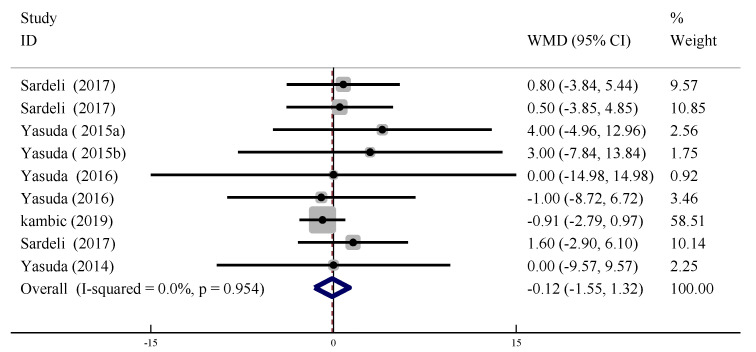
Forest plot of the effect of LL-BFR training on resting DBP.

**Figure 10 ijerph-19-06750-f010:**
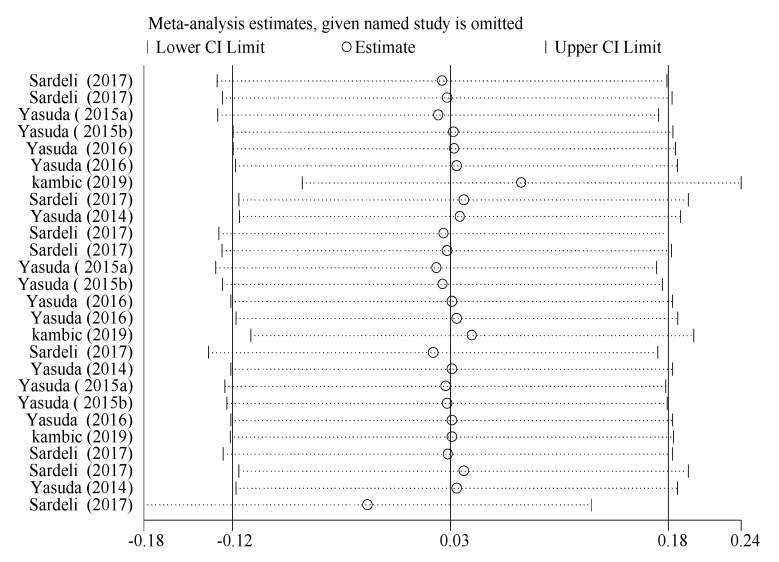
Sensitivity analysis for resting hemodynamic outcomes.

**Figure 11 ijerph-19-06750-f011:**
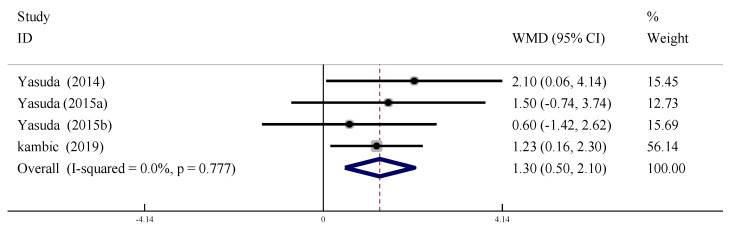
Meta-analysis of the effect of LL-BFR training on FMD of the older adults.

**Figure 12 ijerph-19-06750-f012:**
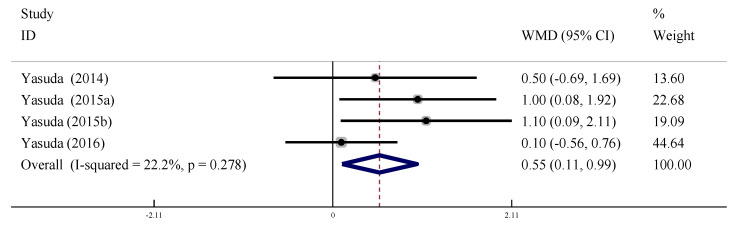
Meta-analysis of the effect of LL-BFR training on CAVI of the older adults.

**Figure 13 ijerph-19-06750-f013:**
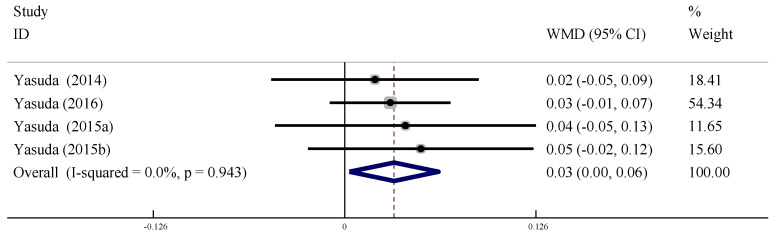
Meta-analysis of the effect of LL-BFR training on ABI of the older adults.

**Figure 14 ijerph-19-06750-f014:**
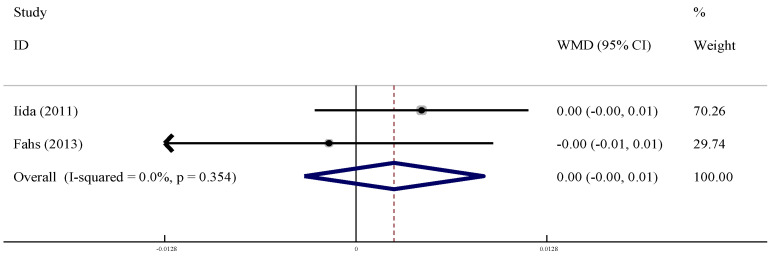
Meta-analysis of the effect of LL-BFR training on VC in the older adults.

**Table 1 ijerph-19-06750-t001:** Characteristics of participants, training protocol and BFR in studies included.

Author and Year	Group	Age	N	The Characteristics of the Exercise of the Experimental Group				The Method of Blood Flow Restriction		Measurement
				Protocol	Duration/Frequency	Volume	Exercise Mode	Cuff Pressure	Cuff Width	
Sardeli [15] (2017)	LL-BFR	64.3 ± 5.04	21	BFR-RT (30% 1RM)	One-time training	75 reps (30 + 3 × 15)	Leg press	100% arterial pressure	17.5 cm	HR; SBP; DBP
	LL-training		21	RT (30% 1RM)		4 sets until failure				
	HL-training		21	RT (80% 1RM)						
	CON		21	Daily activity		10 min	Seated in leg press machine			
Yasuda [16] (2015a)	LL-BFR	72 ± 6	9	BFR-RT (30% 1RM)	12 wk; 2 days/wk	75 reps (30 + 3 × 15)	Elastic band of arm curl exercise	180–270 mmHg 196 ± 18 mmHg	3 cm	HR; SBP; DBP; CAVI; ABI; FMD
	LL-training	68 ± 5	8	RT (30% 1RM)						
Yasuda [17] (2015b)	LL-BFR	72 ± 7	7	BFR-RT (30% 1RM)	12 wk; 2 days/wk	75 reps (30 + 3 × 15)	Elastic band of bilateral arm curl and triceps press down	230–270 mmHg 202 ± 8 mmHg	3 cm	HR; SBP; DBP; CAVI; ABI; FMD
	LL-training	67 ± 6	7	RT (30% 1RM)						
Yasuda [8] (2014)	LL-BFR	71 ± 7	9	BFR-RT (20–30% 1RM)	12 wk; 2 days/wk	75 reps (30, 20, 15, 10)	knee extension leg press	120–270 mmHg	5 cm	HR; SBP; DBP; CAVI; ABI; FMD
	CON	68 ± 6	10	Daily exercise						
Yasuda [18] (2016)	LL-BFR	70 ± 6	10	BFR-RT (35–45% 1RM)	12 wk; 2 days/wk	75 reps (30 + 3 × 15)	Elastic band of bilateral squat; knee extension	160–200 mmHg 161 ± 12 mmHg	5 cm	HR; SBP; DBP; CAVI; ABI
	HL-training	72 ± 7	10	RT (70–90% 1RM)						
Shimizu [6] (2016)	LL-BFR	72 ± 4	20	BFR- RT (20% 1RM)	4 wk; 3 days/wk	3 × 20 reps	leg extension; leg press	134 ± 16 mmHg	10 cm	HR; SBP; DBP
	LI-training	70 ± 4	20	RT (20% 1RM)		3 × 20 reps				
Staunton [19] (2015)	LL-BFR	69 ± 1	13	BFR-RT (20% 1RM)	One-time training	75 reps (30 + 3 × 15)	leg press	121 ± 4 mmHg	10.5 cm	HR; SBP; DBP
	LI-training		13	RT (20% 1RM)						
	LL-BFR		13	BFR-WT (4 km/h)		4 × 2 min	walking without BFR	126 ± 5 mmHg	10.5 cm	
	LI-training		13	WT (4 km/h)						
Pinto [20] (2018)	LL-BFR	67 ± 1.7	18	BFR-RT (20% 1RM)	One-time training	3 × 10 reps	knee extension	143.7 ± 4.8 mmHg	18 cm	HR; SBP; DBP
	HL-training		18	RT (65% 1RM)						
Fahs [21] (2013)	LL-BFR	55 ± 7	16	BFR-RT (30% 1RM)	6 wk; 3 days/wk	3 × 30 reps	knee extension	150–240 mmHg	5 cm	VC
	LI- training		16	RT (30% 1RM)		3 × 30 reps				
Iida [9] (2011)	LL-BFR	67.4 ± 1.6	9	BFR-WT (67 m/min)	6 wk; 5 days/wk	20 min	walking with and without BFR	140–200 mmHg	NS	VC
	LI- training	68.7 ± 2.8	7	WT (NS)		20 min				
Kambic [22] (2019)	LL-BFR	64.9 ± 1.6	12	BFR-RT (30–40% 1RM)	8 wk; 2 days/wk	30 reps + 45 min	knee extension with BFR; aerobic exercise training with and without BFR	145–150 mmHg	23 cm	HR; SBP; DBP; FMD
	LI-training	56.2 ± 6.5	12	usual exercise routine		45 min				
Pinto [23] (2016)	LL-BFR	57 ± 7	12	BFR-RT (20% 1RM)	One-time training	3 × 15 reps	leg-press	195.8 ± 19.7	18 cm	HR; SBP; DBP
	HL-training		12	RT (65% 1RM)						

BFR-RT, blood flow restriction combined with resistance training; BFR-WT, blood flow restriction combined with walking training; NS, not reported; HR, heart rate; SBP, systolic blood pressure; DBP, diastolic blood pressure; FMD, flow-mediated dilation; CAVI, cardio ankle vascular indexes; ABI, ankle brachial indexes; VC, venous compliance; wk, week; reps, repetitions.

**Table 2 ijerph-19-06750-t002:** Results of meta-regression analysis on acute hemodynamic response outcomes.

Variable	Coef.	Std.Err.	t	*p* > |t|	[95%CI]	
Control group	−0.4852251	0.3340651	−1.45	0.159	−1.172345	0.2027948
BFR cuff pressure	−0.0286606	0.0111422	−2.57	0.016	−0.0516085	−0.0057128
BFR cuff width	0.1884616	0.0718443	−2.62	0.015	−0.3364277	−0.0404955
Exercise volume	0.0513778	0.01922	−2.67	0.013	−0.0909621	−0.0117936
_cons	12.26184	3.431454	3.57	0.001	5.194628	19.32905

**Table 3 ijerph-19-06750-t003:** Subgroup analysis results of regulatory variables for acute cardiovascular response outcomes.

Variable	BFR-RT vs. LL-RT				BFR-RT vs. HL-RT			
	No. of Trials	SMD (95% CI)	*I*^2^/%	*p*	No. of Trials	SMD (95% CI)	*I*^2^/%	*p*
HR	5	1.071 (0.15, 1.99)	86.2	0.023	3	0.120 (−0.27, 0.51)	0	0.545
SBP	5	1.339 (0.27, 2.41)	88.7	0.014	3	1.163 (−0.37, 2.70)	91.3	0.137
DBP	5	1.154 (0.12, 2.18)	88.4	0.028	3	1.694 (−0.55, 3.93)	95.1	0.138
120~135 mmHg; 17.5 cm; 75 Reps	6	1.406 (0.25, 2.56)	92.2	<0.05	3	0.214 (−0.14, 0.57)	0	0.231
140~150 mmHg, 18 cm; 30 Reps	6	1.039 (0.11, 1.97)	84.3	<0.05	3	2.598 (−0.14, 5.34)	96	0.063
190~200 mmHg, 18 cm; 45 Reps	3	0.794 (0.31, 1.28)	0	0.351	3	0.253 (−0.21, 0.72)	0	0.284

**Table 4 ijerph-19-06750-t004:** Results of meta-regression analysis on resting hemodynamic outcomes.

Variable	Coef.	Std.Err.	t	*p* > |t|	[95%CI]	
Control group	0.1683382	0.2674929	0.63	0.535	−0.38374	0.7204163
BFR cuff pressure	−0.0020041	0.0077875	−0.26	0.799	−0.0180766	0.0140684
BFR cuff width	−0.0692056	0.0393233	−1.76	0.091	−0.150365	0.0119538
Exercise cycle	−0.0668353	0.0635976	−1.05	0.304	−0.1980944	0.0644237
_cons	1.296574	1.165262	1.11	0.277	−1.108409	3.701556

**Table 5 ijerph-19-06750-t005:** Results of subgroup analysis on resting hemodynamic outcomes.

Variable	LL-BFR vs. LL-Training				Variable	LL-BFR vs. HL-Training			
	No. of Trials	SMD (95% CI)	*I*^2^*/*%	*p*		No. of Trials	SMD (95% CI)	*I*^2^*/*%	*p*
HR	4	0.00 (−0.40, 0.40)	0	0.81	HR	2	0.04 (−0.46, 0.54)	0	0.87
SBP	4	−0.58 (−1.77, 0.61)	85.4	0.34	SBP	3	0.02 (−0.41, 0.45)	0	0.92
DBP	4	−0.05 (−0.35, 0.45)	0	0.99	DBP	3	0.03 (−0.41, 0.46)	0	0.91
BFR training cycle; cuff pressure; cuff width									
Single session; ≤130 mmHg; 17.5 cm	3	0.01 (−0.34, 0.36)	0	0.96	Single session; ≤130 mmHg; 17.5 cm	3	0.10 (−0.26, 0.44)	0	0.60
8 weeks; 150~170 mmHg; 23 cm	3	−0.99 (−2.47, 0.49)	87.5	0.19	12 weeks; 150~170 mmHg; 5 cm	5	−0.06 (−0.45, 0.34)	0	0.78
12 weeks; 180~210 mmHg; 5 cm	6	0.23 (−0.18, 0.64)	0	0.27					

## Data Availability

The datasets generated and analyzed during the current systematic review and meta-analysis are available from the corresponding author on reasonable request.

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
