# Peer review of "Effects of Low-Load Blood Flow Restriction Training on Hemodynamic Responses and Vascular Function in Older Adults: A Meta-Analysis"

_ijerph, 2022, doi:10.3390/ijerph19116750_

Round 1

Reviewer 1 Report

With their manuscript entitled "Effects of Low-load Blood Flow Restriction Training on Hemodynamic Responses and Vascular Function in Older Adults: A Systematic Review and Meta-analysis", Zhang, Tian and Wang provide a comprehensive meta-analysis of the available literature on how low-load training with blood flow restriction impacts hemodynamic and cardiovascular parameters in people/patients of 50 years or older. This topic is of interest for a broad audience of researchers and clinicians.

Minor points:

1) All non-standard abbreviations should be explained in the abstract and at their first mention in the text, especially as this journal is considered multi-disciplinary. In the abstract that would include FMD, ABI, and CAVI. 

2) Line 47: I am wondering if this should be "KAATSU" instead of "KAASU" (without a "T"). 

3) I think the nature of this article is a meta-analysis, and not so much a review article. I would recommend to change the title accordingly. 

4) Although there is an incredible amount of literature related to endothelial progenitor cells (EPC) available, recent high-profile research has shown that this cell type does not exist. The most notable scientists in the field today agree. What has been demonstrated is that the bone marrow-derived cells that were previously termed "EPCs" are actually hematopoietic progenitor cells, which have long been known to express non-specific EC markers like CD31. This would explain why nobody has ever been able to show that a circulating EPC actually gives rise to an EC in any tissue (they have been detected in the perivascular space etc.). To make a long story short: please delete lines 401 to 405 (starting with "This study further..."). 

5) I think it would be highly appreciated by any clinicians (or clinical scientists), if the authors could give a recommendation on the most beneficial LL-BFR training protocol. This should be based on their meta-analysis, but can include personal preference etc. 

Author Response

Dear reviewer,

Thank you very much for your time involved in reviewing the manuscript and your very encouraging comments on the merits.

We have carefully reviewed the comments and revised the manuscript accordingly. Hope the explanation has fully addressed all of your concerns. Point-by-point response to reviewer are attached below this letter.

Reviewer 2 Report

Manuscript of Zhang et al. entitled: “Effects of Low-load Blood Flow Restriction Training on Hemodynamic Responses and Vascular Function in Older Adults: A Systematic Review and Meta-analysis” attempted to reveal how a specific training regimen that combines low intensity exercise with blood flow occlusion (LL-BFR) influences hemodynamic response and vascular function in older adults. It has been shown by others that resistance training preserves muscle performance throughout the aging process, and thus has also a positive effect on cardiovascular health of the older adults. Therefore, especially high-load resistance training (HL-RT) is recommended for older people, because low-load resistance training (LL-RT) is much less effective. It has to be mentioned that HL-RT is often too difficult to be performed by even healthy older people, therefore, LL-BFR may be consider as an attractive alternative training method. If LL-BRF is to became a widely used training method for reducing the risk of cardiovascular disease, it is necessary to understand all physiological impacts of this promising training program on cardiovascular performance. Reading the manuscript and appreciating the arguments, I identified the following major and minor issues which need to be addressed by the authors. In summary, the manuscript is promising and authors should be encouraged to revise it.

Major issue

  1. Authors selected data on elderly with age of more than 50 years. However, no information about their BMI was mentioned. Could this parameter affect outcomes of LL-BRF? It should be discussed by authors.

Minor issues

  1. Pinto et al. (2015) was referenced in Figures 3,4,5 and 6; however, I found only Pinto et al. (2016) and (2018) in the list of references. It seems to me that it is mistake and should be corrected.
  2. In all manuscript, a space has to be added between the numerical value and unit symbol.
  3. Figure 2 was cropped and some parts of figure are missing.

Author Response

(The authors gave the same response as above.)

Reviewer 3 Report

The abstracat:

The combination of low-load (LL) training with blood
flow restriction (BFR) has re-cently been shown to triggers a series of
hemodynamic response and promote vascular function in various populations. To
date, however, evidence is sparse on how this training regimen influences
hemodynamic response and vascular function in older adults.
Objective: To systematically evaluate the effects of LL-BFR training on
hemodynamic response and vascular function in older adults.
Methods: A PRISMA-compliant systematic review and meta-analysis was
conducted. The sys-tematic literature research was performed in the following
electronic databases from inception to 30 February 2022: PubMed, Web of
Science, Scopus, EBSCO host, Cochrane Library and CNKI. Subsequently, a
meta-analysis with inverse variance weighting was conducted.
Results: A total of 1437 articles were screened, and 12 randomized controlled
trials with a total 378 subjects were included in the meta-analysis. The
Meta-analysis results showed that: LL-BFR training caused a significant acute
increase in heart rate (WMD: 4.02, 95% CI: 0.93, 7.10, P < 0.05), systolic
blood pressure (WMD: 5.05, 95% CI: 0.63, 9.48, P < 0.05) and diastolic blood
pressure (WMD: 4.87, 95% CI: 1.37, 8.37, P<0.01). The acute hemodynamic
response induced by LL-BFR training is similar to that elicited by high-load
(HL) training. Training volume, cuff pressure and width are significant
moderators in subgroup and meta-regression analyses. After 30 min of
training, the resting systolic blood pressure significantly decreased (WMD:
-6.595, 95% CI: -8.88, -3.31, P < 0.01) in LL-BFR training group, but resting
hemodynamic was no significant difference compared with common LL and HL
training; Long-term LL-BFR training significantly improved FMD (WMD: 1.30,
95% CI: 0.50, 2.10, P<0.01), CAVI (WMD: 0.55, 95% CI: 0.11, 0.99, P<0.05) and
ABI (WMD: 0.03, 95%CI: 0.00, 0.06, P<0.05) of older adults.
Conclusion: This systematic review and meta-analysis reveals that LL-BFR
training will cause an acute hemodynamic response in older adults, which can
return to normal level 30 min after training and systolic blood pressure
significantly decreased. Furthermore, the beneficial effect of LL-BFR
training on vascular function is to improve FMD, CAVI and ABI of older
adults. However, due to the influence of the quality of the included studies
and the sample size, more high-quality studies are needed to confirm such
issues as BFR pressure and training risk.

is a honest summary of a well designed and implemented study.

I have no major criticisms/suggestions.

For the general readers, I suggest to add and discuss a few more references, i., e.,:

  1. Cvecka J, Tirpakova V, Sedliak M, Kern H, Mayr W, Hamar D. Physical Activity in Elderly. Eur J Transl Myol. 2015 Aug 25;25(4):249-52. doi: 4081/ejtm.2015.5280. eCollection 2015 Aug 24.
  2. Barberi L, Scicchitano BM, Musaro A. Molecular and Cellular Mechanisms of Muscle Aging and Sarcopenia and Effects of Electrical Stimulation in Seniors. Eur J Transl Myol. 2015 Aug 25;25(4):231-6. doi: 10.4081/ejtm.2015.5227. eCollection 2015 Aug 24.
  3. Hamar Universal Linear Motor Driven Leg Press Dynamometer and Concept of Serial Stretch Loading. Eur J Transl Myol. 2015 Aug 25;25(4):215-9. doi: 10.4081/ejtm.2015.5281. eCollection 2015 Aug 24.

Author Response

(The authors gave the same response as above.)
